**Title:** Committenze semantiche: Wikidata per la ricerca storico-artistica

**Author:** Alessio Ionna

**Affiliation:** Università di Macerata

**Keywords:** Wikidata, Committenza artistica, Ricerca storico-artistica, Linked Open Data.

**Author biography:** Laureato magistrale con votazione 110 e lode in Management dei beni culturali presso l'Università degli studi di Macerata, con una tesi in Storia delle immagini sul ciclo decorativo di Palazzo Millo ad Ancona. Nel 2021 questa tesi vince il premio tesi di laurea sui beni vincolati privati, promosso dall'Associazione Dimore Storiche Italiane.
Nel 2023 è stato borsista presso la Fondazione Giorgio Cini, con un progetto di valorizzazione digitale del patrimonio della fondazione veneziana.
Attualmente è dottorando del corso di Umanesimo e tecnologie dell'Università degli studi di Macerata, sotto la supervisione del prof. Pierluigi Feliciati, dove si occupa di descrivere in ambiente digitale semantico il fenomeno della committenza artistica della famiglia Buonaccorsi di Macerata.

**Author bibliography**

- Ionna A. (2022), *La rappresentazione del fenomeno delle committenze artistiche in ambiente digitale: il caso della famiglia Buonaccorsi*, «Digitalia. Rivista del digitale dei Beni Culturali», 17, 2, pp. 133-138, URL: https://digitalia.cultura.gov.it/article/view/2947.
- Ionna A. (2023), *Rappresentazione digitale semantica del contesto storico-culturale dell'isola di San Giorgio Maggiore e delle collezioni artistiche della Fondazione Giorgio Cini di Venezia*, «Il Capitale Culturale. Studies on the Value of Cultural Heritage», 28, pp. 557–573, DOI: https://doi.org/10.13138/2039-2362/3288.

**Abstract:** Negli ultimi anni le scienze umane hanno esplorato le potenzialità delle tecnologie digitali e in particolare le applicazioni del Web semantico per studiare fenomeni culturali complessi come le committenze artistiche. La ricerca intende presentare una possibile applicazione di Wikidata in ambito storico-artistico per rappresentare in un ambiente semantico e collaborativo il mecenatismo di una famiglia nobile, la famiglia Buonaccorsi di Macerata, una delle dinastie più influenti dello Stato Pontificio tra il XVII e XVIII secolo.

Attraverso lo spoglio sistematico delle fonti è stato possibile realizzare in Wikidata un dataset di oltre 400 elementi ascrivibili al contesto della casata maceratese. Ogni elemento relativo alla committenza Buonaccorsi è stato correttamente descritto e relazionato attraverso le proprietà Wikidata ritenute più idonee, permettendo così di restituire una visione diacronica e interconnessa del fenomeno artistico. Inoltre, la possibilità di referenziare le asserzioni ha rinforzato l'affidabilità dei dati, rendendo in questo modo più autorevole il loro riuso in ambito scientifico. Questo studio dimostra come Wikidata, attraverso i Linked Open Data, possa offrire nuove prospettive per la ricerca storico-artistica, raccogliendo in un'unica piattaforma - in forma di dati - fonti disseminate presso diverse sedi, ottimizzando le attività di legate al recupero delle informazioni. Inoltre, evidenzia come le piattaforme aperte possano innovare la ricerca nelle scienze umane, favorendo la costruzione e la diffusione di nuove conoscenze e strumenti di ricerca per gli storici e per le comunità d'interesse.

**Theme:** Reuse of data, Research assessment

**Language of the presentation:** Italian