# OpenReview forum: "Committenze semantiche: Wikidata per la ricerca storico-artistica"
_wikimedia.it/Wikidata_and_Research/2025/Conference — WD&R differentformat_

### Official Review · ~Lucia_Sardo1 · 2025-01-07
**revisione**

**Originality:** 3
**Impact:** 3
**Confidence:** 4

**Review:**

La relazione presenta i risultati di progetto che ha portato al caricamento di un importante dataset relativo alla famiglia maceratese dei Buonaccorsi. L'abstract purtroppo non riporta la metodologia usata per la raccolta dei dati, il loro trattamento e le modalità utilizzare per il caricamento del dataset stesso. Sicuramente progetti di questo tipo rappresentano buone pratiche per l'arricchimento di Wikidata, ma non è possibile valutare la presenza di aspetti innovativi o l'originalità del progetto con le informazioni a disposizione.

**Compliance:**

4

**Scientific Quality:**

4

---

### Official Review · ~Alessandra_Boccone1 · 2025-01-07
**Buona pratica nell'uso di WD nell'ambito storico-artistico**

**Originality:** 4
**Impact:** 4
**Confidence:** 5

**Review:**

L'attività illustrata nell'abstract rappresenta un valido esempio di buone pratiche relative all'uso di Wikidata nell'ambito della ricerca storico-artistica, attraverso un uso consapevole delle fonti e con una sicura ricaduta positiva sulla ricerca in tale ambito.
Sarebbe stato utile descrivere brevemente nell'abstract il flusso di lavoro e le eventuali criticità rilevate, per avere un'idea più completa del progetto e maggiori elementi di valutazione.

**Compliance:**

4

**Scientific Quality:**

4

---

### Decision · Program_Chairs · 2025-02-05

**Decision:**

Accept (LT)

**Comment:**

Dear Author,
thank you very much for your proposal. We regret to inform you that your proposal was not selected among the papers.

Even if not selected as paper, we consider your proposal relevant and interesting and we would like to propose you to prepare instead a lightening talk (if you - or another member of your team - can participate in presence at the conference) or a poster (which can be exhibited even if you will not attend the conference).

It would be a pleasure to learn more about your work through a lightening talk or a poster.
Thank you for submitting a proposal and please let us know if you like the idea of converting it into a lightening talk or a poster and which format you prefer.

Regards,
The scientific committee of the conference Wikidata and Research